# Immune Cells, Gut Microbiota, and Vaccines: A Gender Perspective

**DOI:** 10.3390/cells13060526

**Published:** 2024-03-17

**Authors:** Pierluigi Rio, Mario Caldarelli, Monica Chiantore, Francesca Ocarino, Marcello Candelli, Antonio Gasbarrini, Giovanni Gambassi, Rossella Cianci

**Affiliations:** 1Department of Translational Medicine and Surgery, Catholic University of Rome, Fondazione Policlinico Universitario A. Gemelli, IRCCS, 00168 Rome, Italy; pierluigi.rio18@gmail.com (P.R.); mario.caldarelli01@icatt.it (M.C.); mchiantore93@gmail.com (M.C.); francesca.ocarino01@icatt.it (F.O.); antonio.gasbarrini@unicatt.it (A.G.); giovanni.gambassi@unicatt.it (G.G.); 2Department of Emergency, Anesthesiological and Reanimation Sciences, Catholic University of Rome, Fondazione Policlinico Universitario A. Gemelli, IRCCS, 00168 Rome, Italy; marcello.candelli@policlinicogemelli.it

**Keywords:** vaccines, gender, hormones, immune cells, gut microbiota

## Abstract

The development of preventive and therapeutic vaccines has played a crucial role in preventing infections and treating chronic and non-communicable diseases, respectively. For a long time, the influence of sex differences on modifying health and disease has not been addressed in clinical and preclinical studies. The interaction of genetic, epigenetic, and hormonal factors plays a role in the sex-related differences in the epidemiology of diseases, clinical manifestations, and the response to treatment. Moreover, sex is one of the leading factors influencing the gut microbiota composition, which could further explain the different predisposition to diseases in men and women. In the same way, differences between sexes occur also in the immune response to vaccines. This narrative review aims to highlight these differences, focusing on the immune response to vaccines. Comparative data about immune responses, vaccine effectiveness, and side effects are reviewed. Hence, the intricate interplay between sex, immunity, and the gut microbiota will be discussed for its potential role in the response to vaccination. Embracing a sex-oriented perspective in research may improve the efficacy of the immune response and allow the design of tailored vaccine schedules.

## 1. Introduction

The introduction of vaccines has changed the course of medical history, allowing the prevention of severe infectious diseases and reducing their clinical and economic burden on human health [1].

According to the World Health Organization (WHO), vaccines are “biological preparations that enhance immunity to a particular disease” [2]. Each year, they prevent the deaths of about 3.5–5 million individuals [3], and it is estimated that more than 50 million deaths worldwide can be prevented through vaccination between 2021 and 2030 [4].

Vaccination campaigns led to the complete eradication of smallpox about 40 years ago [5]; other diseases, such as poliomyelitis, will hopefully be eradicated in the future [6].

Vaccines belong to the field of preventive medicine since they are administered to healthy individuals as a first line of defense against infections, but they also provide “herd immunity” within a population by reducing the number of susceptible people thus interrupting the chain of transmission [7].

There is a significant variation in the immune response to vaccination among humans, depending on intrinsic, extrinsic, behavioral, nutritional, environmental, and also specific vaccine factors [8,9,10].

Despite the well-known impact of gender on health and disease, the differences between men and women are not systematically analyzed. Indeed, for decades, clinical studies have seldom involved women, and no relevant data about gender-related differences in clinical patterns and therapeutic response have been gathered [11].

In the last few years, as advocated by an Editorial published in *Nature*, researchers have begun to “put gender on the agenda” [12].

The novel definition of “gender medicine” is not the study of sex-specific diseases, mostly regarding the reproductive systems, but the evaluation of the genetic, epidemiologic, clinical, therapeutic, prognostic, and psychological differences in diseases involving both men and women [13].

The immune system elicits different responses to antigens in men and women [14]. Consistently, there is a substantial difference in the prevalence of autoimmune diseases, as almost 80% of patients are women [15].

Modern research has been progressively clarifying the immunomodulatory role of the gut microbiota and its influence on vaccine immunogenicity [16]. Several studies have identified sex differences in gut microbiota composition [17] that could further explain the dimorphisms in the immune responses of the two sexes.

Our narrative review aims to shed light on the immunobiological variations between men and women and evaluate how gender-related differences may influence the outcome of vaccination, with a focus on the potential impact of sex-specific gut microbiota on vaccine immune response.

## 2. Sex Differences in Health and Disease

Gender medicine is a branch of medicine that studies anatomical, physiological, biological, functional, and social differences between men and women and the influence of these factors on health, disease, and responses to therapies. It describes the differences in symptoms, clinical development and drug response, as well as preventive approaches between men and women diagnosed with the same disease [18].

The first time that the concept of “gender medicine” was conceived was in 1991, when a cardiologist, Bernardine Healy, highlighted the discrimination of women in the management of cardiovascular disease in an article published in *The New England Journal of Medicine*. Most specifically, she was documenting that the vast majority of research on ischemic heart disease was exclusively performed in men [19]. Following Healy’s article, increased efforts have been made to include women in medical studies and promote sex-specific medicine.

According to the WHO, sex refers to the biological characteristics that define humans as men or women, whereas gender refers to the characteristics of women, men, girls, and boys that are socially constructed and includes roles, behaviors, activities, and attributes that a given society considers appropriate for the two different sexes [20]. Therefore, gender medicine deals with the impact of sex and gender on the physiology, pathophysiology, and clinical features of a disease in order to achieve evidence-based therapeutic decisions for both men and women [21,22] (Figure 1).

Men and women differ primarily due to genetic sex determination at conception. This occurs when an oocyte fuses with a sperm cell carrying either an X or a Y chromosome, resulting in a zygote with XX chromosomes (female sex) or XY chromosomes (male sex). Sexual differentiation begins during intrauterine life and is contingent upon the development of the testis, enabled by the Sex-determining Region Y (SRY) gene in the Y chromosome. In the absence of SRY and with the presence of two X chromosomes, the interplay of various genes dictates ovarian development. Once the gonads differentiate as testis or ovaries, their hormonal secretions determine the sexual phenotype of the body [23].

The Genotype Tissue Expression Project (GTEx), launched in 2010 and aimed at understanding gene expression phenomena, has examined 44 tissue types belonging to 838 subjects and documented considerable differences in gene expression between men and women in every tissue, including those from organs without sexual characterization. The differences expressed are small but are present to a considerable extent in both X-linked and autosomal genes. Thus, sex seems to influence gene expression and genetic regulation across tissues [24].

Sex hormones themselves directly influence disease susceptibility and clinical presentation. In chronic kidney disease, men progress faster to end-stage renal disease partly due to male hormones increasing oxidative stress, activating the renin–angiotensin system and exacerbating fibrosis in the injured kidney. In contrast, female hormones appear to exhibit a renoprotective effect [25]. In cardiovascular diseases, the expression of elevated levels of estrogen receptors in the heart induces a T helper 2 (Th2) response, leading to inflammatory effects involving both the endothelium and the immune cells [26]. This phenomenon is now considered to be playing a role in the pathogenesis of heart failure with preserved ejection fraction, which is observed much more in women than in men [27].

Regarding inflammation-mediated medical conditions, there are notable differences in epidemiology and clinical presentation between men and women. For example, in the context of acute pneumonia, women seem to exhibit a more robust inflammatory response, leading to a more favorable outcome compared to men. Due to their two X chromosomes, one undergoing random inactivation in cells, women emerge as a cellular mosaic as they integrate genes from both maternal and paternal X chromosomes. Their genetic makeup carries more X-linked genes that code for proteins involved in the immune response, such as Toll-like receptor (TLR) family proteins and nuclear factor kappa-light-chain-enhancer of activated B cells (NF-κB), providing important assets in fighting infections [28]. On the other hand, women’s stronger immune reactivity may trigger the development of autoimmune diseases, whose prevalence is much higher compared to men [29]. The predisposition towards autoimmunity in women is influenced not only by the X chromosome but also by the vast range of effects that sex hormones exert on the molecular mechanisms of the innate and adaptive immune system [30]. For instance, estradiol enhances the NF-κB pathway and positively correlates with interleukin (IL)-6 levels in systemic lupus erythematosus [31]. Additionally, estradiol increases B cell activating factor (BAFF) protein transcription leading to the dysregulation of thyroid function (e.g., Graves’ disease) [32].

Men and women also exhibit different responses to treatment due to distinct drug pharmacokinetics and pharmacodynamics [33]. For instance, given that women typically have a lower body weight and a higher proportion of adipose tissue compared to men, the identification of pharmacokinetic variances is quite common. The leading role of estrogens in determining the sex-specific distribution of adipose tissue has been described, together with their metabolic effects, such as the promotion of insulin sensitivity and glucose uptake and protection against diabetes and obesity [34]. Since the adipose tissue acts as an endocrine organ and produces inflammatory cytokines, it contributes to immune dysregulation and to the reduced vaccine responses observed in obese individuals [35,36].

## 3. Sex Differences in Immunity

### 3.1. Innate Immunity

The innate immune system is constituted by various physical, biochemical, and cellular mechanisms that act as the initial defense against pathogens [37]. The cellular components of the innate immune system are represented by monocytes, macrophages, neutrophils, dendritic cells (DCs), natural killer (NK) cells, eosinophils, and basophils [37].

In humans, sexual dimorphism is evident in various immune processes, such as in response to pathogens and vaccines. In a study involving 534 healthy individuals, the ex vivo inflammatory response by monocytes activated by multiple microbial stimuli was influenced by factors such as sex, age, and season of the year [38]. Women generally exhibit lower rates of infection across a spectrum of bacterial, viral, and parasitic infections [39], with only a few exceptions (e.g., higher incidence rates for pertussis in women [40]).

Steroid hormones like estrogen and progesterone in females, as well as testosterone and other androgens in males, are responsible for regulating diverse biological processes. These hormones play a role in modulating various aspects of the innate immune system [41]. Estrogen, progesterone, and testosterone interact with nuclear hormone receptors, such as the estrogen receptor (ER), progesterone receptor (PR), and androgen receptor (AR), respectively, in a broad array of cell types, including immune cells. When these receptors bind to their respective ligands, they exhibit a strong affinity for specific DNA sequences called hormone response elements (HREs) situated in the promoters of target genes [41].

#### 3.1.1. Estrogens

The role of estrogens in the regulation of the development and functioning of the female reproductive system has been extensively studied [42].

Four forms of endogenous estrogen, namely estrone (E1), estradiol (E2), estriol (E3), and etestrol (E4), are responsible for hormonal effects [43]. These effects are predominantly executed through their binding to estrogen receptors (ERs) via two distinct mechanisms. The classical mechanism involves the entry of estrogen into the cell and the binding to the ER in the nucleus, with a consequent activation or repression of specific genes [44]. Alternatively, estrogens can rapidly stimulate cells through “nongenomic” mechanisms by binding to estrogen receptors (ERs) located on the plasma membrane or on the endoplasmic reticulum. This prompts immediate responses, such as alterations in Ca^2+^ levels or kinase activity [45].

ER signaling plays a crucial role in regulating various aspects of cellular processes, including cell proliferation, growth factors, cytokines (e.g., interferons, IL-6, IL-1, vascular endothelial growth factor (VEGF), amphiregulin, and transforming growth factor (TGF)-beta), receptors and signaling pathways (e.g., NF-kB, signal transducer and activator of transcription (STAT), TGF-beta, and tumor necrosis factor (TNF)), as well as transcription factors and coregulators (e.g., c-Fos, c-Myc, Myb, and JunB). The transcriptional regulation by ER extends to immune cell functions, and significant fluctuations in estrogen concentrations throughout a woman’s life course can lead to alterations in the activation of ER signaling in immune cells [14]. E2 can bind to both intracellular (genomic) and membrane-bound (nongenomic) estrogen receptors, leading to transcriptional changes in immune cells.

The signaling of estrogen receptors governs the immediate inflammatory and innate immune reaction of neutrophils. Experiments conducted both in vivo and in vitro demonstrate that levels of estrogen impact the expression of ERα and ERβ in neutrophils [46]. In humans, neutrophils extracted from premenopausal women during the follicular phase exhibit a higher expression of ERα and ERβ compared to neutrophils isolated during the ovulatory phase of the menstrual cycle [47]. In rats, administering E2 or selective ER agonists (such as 4,4′,4″-(4-propyl-[1H]-pyrazole-1,3,5-triyl) trisphenol [PPT] and 2,3-bis(4-hydroxyphenyl)-propionitrile [DPN]) induces an increase in the expression of genes associated with inflammation and extracellular matrix remodeling. These include genes encoding 12-lipoxygenase, fibulin-1, furin, and calgranulin B [48].

In the case of an infection with *Pseudomonas aeruginosa*, the signaling pathways involving both ERα and ERβ hinder the ability of neutrophils to contrast the infection in female mice, marking a stark difference to the response observed in male mice [49].

Data currently available demonstrate that E2 (estradiol) contributes to an increase in neutrophils in infected tissues. However, the specific mechanism underlying this phenomenon is not well explored in most studies. It remains unclear whether the rise in neutrophils is a result of infected cells producing attracting chemokines or if ER signaling within neutrophils is responsible for these effects [14].

Natural killer (NK) cells express both Erα and Erβ, and the modulation of NK cell activity occurs through signaling of these receptors [50]. Treatment with E2 leads to an increase in the number of NK cells; however, it concurrently reduces cytotoxicity, partially achieved by altering the expression of genes associated with cellular cytotoxicity and proliferative activity. These include genes encoding CD94 and IFN-gamma [14]. In postmenopausal women, the administration of combined estrogen and medroxyprogesterone hormone replacement therapy is linked to a decrease in NK cell cytotoxicity, as well as to a reduction in the synthesis of IL-2 and IFN-gamma [51].

Both ERα and ERβ receptors, in addition to the membrane-bound G-protein-coupled receptor (GPR30/GPER-1), are expressed in macrophages [52]. ER signaling influences NF-kB nuclear translocation specifically through ERα, not ERβ. This modulation occurs by impairing the transcriptional activity of p65 and preventing NF-kB intracellular localization during an LPS-induced inflammatory response [53]. GPR30/GPER-1 exerts anti-inflammatory effects in macrophages by regulating TLR4 at the cell surface [52].

ER signaling plays a role in shaping sex differences in disease outcomes. For instance, estrogen (E2) enhances IL-4-induced M2 gene expression in bone-marrow-derived and alveolar macrophages in female mice after respiratory allergen challenge. Additionally, in a cutaneous wound repair model, E2, through ERα signaling, diminishes proinflammatory gene expression in macrophages, thereby promoting the process of wound healing [14].

Dendritic cells (DCs) and progenitor subsets express both ERα and ERβ, and the modulation of DC differentiation and activity occurs through these receptors [14]. Studies conducted both in vitro and in vivo demonstrate that ER signaling, particularly through ERα, is crucial for regulating differentiation, cytokine production, and activity. As an example, E2 signaling through ERα, but not ERβ, facilitates the differentiation of DCs stimulated by GM-CSF (granulocyte-macrophage colony-stimulating factor) [54]. Studies conducted in vivo using mice deficient in either ERα or ERβ reveal that the activation of ERα, dependent on estrogen, is essential for DC differentiation from bone marrow and the production of cytokines [55].

Both ERα and ERβ are expressed in T cells. Generally, CD4+ T cells exhibit higher levels of ERα expression compared to ERβ, whereas CD8+ T cells express similar levels of both receptors [14]. In mouse models of autoimmune diseases, the signaling of ERα and ERβ often performs opposing functions. Specifically, ERα signaling tends to be proinflammatory, while ERβ signaling tends to be anti-inflammatory, particularly in CD4+ T cells [56]. In both mice and humans, signaling through ERβ in T cells tends to suppress inflammatory T cell responses. Individuals with Crohn’s disease, ulcerative colitis, or inflammatory bowel disease show a reduced expression of ERβ in T cells, both within the ileal mucosa and in the peripheral tissues [57].

The ERα, involved in B cell activity and antibody production, has DNA binding sites in the enhancer hs1.2, which is a regulatory domain in the immunoglobulin (Ig) heavy (H) chain locus. Interestingly, a genetic variant in the *2 allele in hs1.2 has been shown to confer protection to women who exhibit a lower clinical severity of COVID-19 disease [58].

#### 3.1.2. Progesterone

Recent studies have shown the inhibitory effects of progesterone on immune responses, especially in the context of inflammatory processes [59].

Progesterone exerts inhibitory effects on the activation of murine dendritic cells, macrophages, and NK cells [60]. Progesterone plays intriguing immunoregulatory roles by influencing the generation of various immune cell types. Leigh et al. explored the role of progesterone in modulating dendritic cell (DC) function, generating several key findings. First, the capacity of progesterone to reduce inflammatory cytokine production and costimulatory molecule expression in bone marrow-derived DCs depends largely on the TLR that is activated. Progesterone downregulates certain TLR-induced inflammatory mediators through the glucocorticoid receptor (GR) alone and others through both the progesterone receptor (PR) and the GR. Finally, PR agonists sustain the phosphorylation of interferon regulatory factor 3 (IRF-3) following TLR3 activation but not TLR4 activation [61].

In their research on the effects of progesterone on macrophages, Menzies et al. have investigated how this hormone influences their activity through alternative pathways distinct from those involving nitric oxide and interleukin 2. Bone marrow cells isolated and differentiated from male BALB/c mice were subjected to varying concentrations of progesterone and stimulated with lipopolysaccharide (LPS) for innate activation, IL-4 for alternative activation, or with a combination of LPS and IL-4. This study has found that progesterone reduces not only the activity of inducible nitric oxide synthase 2 in macrophages but also the arginase activity in a dose-dependent manner, irrespective of the stimuli, whether induced by LPS (innate activation), IL-4 (alternative activation), or a combination of LPS and IL-4. Additionally, progesterone’s ability to decrease IL-4-induced cell surface expression of the mannose receptor suggests a negative regulation of alternative macrophage activation by this hormone [62].

Recent research indicates that a disruption of the interaction between progesterone and dendritic cells can lead to the reduced production of CD4^+^ T regulatory cells. This association is linked to challenges in placentation and intrauterine growth restriction in mice [63]. Shah et al. proposed some mechanisms of P4-regulated gene transcription to modulate T cell function. Conventionally, extranuclear progesterone receptors (nPRs) remain inactive until binding with P4, forming a dimer that translocates to the nucleus. In the nucleus, it binds to progesterone response element (PRE) sequences within gene promoter regions, thereby altering their transcriptional activity. Alternatively, as a monomer, nPR-P4 acts through the Src kinase to activate the MAPK cascade. P4 bound to membrane progesterone receptors (mPRs) modifies gene transcription regulated by second messengers (cAMP and Ca^2+^) and their associated extranuclear kinases (PKA and PKC) via the MAPK signal transduction cascade. This process results in the phosphorylation of nuclear transcription factors (TFs).

Membrane-bound progesterone receptors (mPRs) and progesterone receptor membrane components (PGRMCs) likely influence T cell receptor (TCR) signal transduction by modulating the activities of MAPKs through Zap70. Additionally, they impact Ca^2+^ mobilization induced by the phospholipase Cγ (PLCγ)-driven production of diacylglycerol (DAG) and 1,4,5-trisphosphate (IP3). These events collectively lead to the modulation of pro-inflammatory gene expression and T cell activation through transcription factors NF-κB, AP-1, and NFAT [64].

Many of the immunological effects attributed to progesterone are mediated by a downstream factor known as the progesterone-induced blocking factor (PIBF). The significance of PIBF in immunoregulation during pregnancy is underscored by recent research, revealing that PIBF-deficient mice exhibit increased decidual and peripheral NK activity. In addition, T cell activation genes are downregulated in CD4^+^ T cells and upregulated in CD8^+^ T cells, leading to the differentiation of T cells into T helper 1 (Th1) cells. Interestingly, PIBF-deficient mice demonstrate lower implantation rates and higher rates of fetal loss compared to mice with intact PIBF activity [59].

#### 3.1.3. Androgens

The term “androgen” refers to any steroid hormone that exhibits masculinizing effects [65]. The biological actions of androgens, which include testosterone, dihydrotestosterone (DHT), as well as androstenedione, dehydroepiandrosterone (DHEA), and its sulfated form (DHEA-S), are typically mediated through the androgen receptor (AR). The androgen receptor functions as a ligand-dependent nuclear transcription factor [66]. The AR can directly impact immune cells by influencing the transcription of immune-regulatory genes [66].

There are two distinct routes of androgen signaling: the conventional or genomic pathway and the unconventional or non-genomic pathway [67]. When androgens are not present, the androgen receptor (AR) is exclusively situated in the cytoplasm and connected to heat-shock proteins (HSPs). This interaction with the ligand triggers the separation of AR from HSPs, initiating the subsequent movement of AR into the nucleus [68].

After AR is transported into the nucleus, the ligand-activated AR binds specific DNA regulatory sequences known as androgen response elements (AREs). This transcription factor, activated by the ligand, regulates gene expression by directly binding DNA and recruiting various coregulators to create complexes. These complexes play a crucial role in inducing epigenetic histone modifications and remodeling chromatin at target genetic loci [66]. The activation of the non-genomic or non-classical pathway results in the swift, transcription-independent effects of androgens, brought about by their interaction with non-classical receptors such as ZIP9 and GPRC6A [67]. Instances of effects triggered by the binding of androgens to non-classical receptors encompass the activation of mitogen-activated kinase (MAPK), protein kinase C (PKC), protein kinase A (PKA), as well as elevations in free intracellular calcium [66].

Androgens can promote neutrophil differentiation and recruitment, leading to an increase in their numbers in both mice and humans [69].

Furthermore, aside from reduced neutrophil counts, global AR knockout (ARKO) mice also exhibited functional defects in neutrophils. While these neutrophils retained normal phagocytic properties, they demonstrated reduced responsiveness to granulocyte-colony stimulating factor-induced proliferation and migratory signals in vitro [66].

Several studies have investigated the impact of androgens on macrophage function, and the overall consensus suggests an immunosuppressive effect [66]. In preclinical models, the administration of the anti-androgen flutamide, after the induction of sepsis, was not only able to restore the low levels of cytokine release by splenic macrophages and splenocytes but also significantly decreased the mortality of post-hemorrhaged mice [70]. Furthermore, testosterone has been observed to reduce the expression of TLR4 in a macrophage cell line, in cultured primary macrophages, and in vivo in mice [71]. In general, testosterone elicits an inhibitory effect on dendritic cells; however, it remains unclear whether this effect is direct or indirect, as the expression of AR by dendritic cells has not been definitively established. In this context, a study conducted in mice revealed that bone marrow-derived dendritic cells (BMDCs) express ER but not AR [72]. In men with hypogonadism, there was an observed increase in the most widespread dendritic cell subset, and this increase was subsequently reversed with testosterone treatment [73].

### 3.2. Adaptive Immunity

The adaptive immune response matures at a later stage and is tailored to the specific pathogen, which remains for a long period and establishes immunological memory [74].

Estradiol and progesterone appear to have opposite effects on activated CD4^+^ T cells. Estradiol was found to enhance the proportion of activated cells and the secretion of immune- and inflammation-related proteins, particularly in low-activated cells. In contrast, progesterone consistently reduced these effects, despite the level of cell activation [75].

As for androgens, when administered to mice, testosterone exerts a negative regulatory effect on the differentiation of Th1 cells by inhibiting IL-12-induced STAT4 phosphorylation. In this mechanism, the androgen receptor (AR) binds to the phosphatase Ptpn1 (protein tyrosine phosphatase non-receptor type 1), resulting in the inhibition of IL-12 signaling in CD4+ cells [76]. Furthermore, testosterone directly interacts with androgen receptors (ARs) on CD4^+^ cells, leading to the upregulation of IL-10 expression. IL-10 is an anti-inflammatory cytokine that promotes a Th2-type immune response [77]. Fijak et al. illustrated that the in vitro treatment of naive T cells with testosterone led to an enlargement of rat murine regulatory T cells (Treg cells) exhibiting immunosuppressive activity. Furthermore, in the same study, it was demonstrated that increased testosterone levels during experimental autoimmune orchitis (EAO) in rats led to a notable augmentation in the population of Treg cells, identified as CD4+ CD25+ Foxp3+, as compared to the control group of animals subjected to EAO [78].

It has been recognized that estrogen boosts humoral responses and enhances the differentiation of B cells and the production of immunoglobulins [79]. Estrogen governs the peripheral populations of B cells, influencing tolerance induction [80]. The treatment of splenic cell populations with estrogens resulted in elevated numbers of marginal zone (MZ) B cells, reduced transitional B cells, and a slight increase in follicular B cells [81].

Regarding B lymphocytes, testosterone appears to have an inhibitory effect [77].

Androgen receptors have not been identified in mature B cells; however, they have been detected in both B-cell precursors and bone marrow stromal cells. Thus, testosterone is likely to play a role in the maturation process of B cells [82]. A plausible explanation for this observation is that testosterone may reduce the concentration of B-cell activating factor (BAFF), a crucial survival factor for B cells. Moreover, AR knockout mice and male mice that have undergone gonadectomy exhibit higher concentrations of BAFF compared to individuals with intact reproductive systems [83]. Likewise, when peripheral blood cells are incubated with testosterone, there is a decrease in the concentration of IL-6. This reduction in IL-6 leads to a decrease in the concentration of IgG and IgM antibodies in vitro, and this effect occurs in a dose-dependent manner. Importantly, this modulation of B-lymphocyte activity by testosterone appears to be independent of B-lymphocyte proliferation, suggesting a negative regulatory role of testosterone on B-lymphocyte function [77] (Table 1).

Altfeld et al. illustrated that during acute human immunodeficiency virus 1 (HIV-1) infection, women exhibit superior control of viral replication compared to men. Conversely, women with untreated chronic HIV-1 infection show a more rapid loss of CD4+ T cells. The sex-specific differences in antiviral response could depend on innate immunity, in particular on type 1 IFN responses, linked to sex hormones and genes encoded by the X-chromosome [84].

Furthermore, in SARS-CoV-2 infection, Takahashi et al. found differences in immune response and disease progression between men and women. Men showed higher levels of innate immune cytokines, such as IL-8 and IL-18, and non-classical monocytes. On the other hand, women exhibited significantly stronger T cell activation. Moreover, higher levels of innate immune cytokines in women are linked to worse clinical outcomes compared to men [85].

### 3.3. Autoimmunity

Autoimmune diseases include a variety of conditions where the body’s immune system, reacting to own antigens, leads to the harm or impairment of self-tissues [86]. For the majority of these conditions, there is an evident sex-based difference in the prevalence of autoimmune diseases, with women being more frequently affected compared to men [87].

This increased susceptibility to autoimmune diseases may arise from the action of sex-specific hormones in disease development, along with their distinct roles in altered reproductive states. Progesterone and androgens exert anti-inflammatory and immunosuppressive effects, generally proving beneficial in the context of autoimmune diseases. Furthermore, prolactin, which is elevated during pregnancy, induces pro-inflammatory effects, contributing to a tendency to exacerbate autoimmune diseases [88]. The effects elicited by estrogens are more complex. In women, the circulating levels of estrogens undergo significant changes in relation to reproductive function. This is crucial because exposure to different levels of estrogens has a profound impact on immunity. At elevated concentrations, such as during pregnancy, estrogen inhibits the Th1 pro-inflammatory pathways, including TNF-a, IL-1b, and IL-6, while simultaneously promoting Th2 anti-inflammatory pathways, including IL-4, IL-10, and TGF-beta. Conversely, at lower levels, as observed during post-menopause, estrogen stimulates pro-inflammatory pathways, including TNF-a and IL-1b [89]. The effects of estrogen outlined earlier could result in either enhanced cell-mediated disease or exacerbated antibody-mediated disease [88].

Some genetic factors may contribute to sex-specific autoimmune disease development, and these might include susceptibility genes, chromosomal distinctions, or epigenetics. The interplay between genetic factors and environmental influences can affect the heritability of autoimmune diseases. Epigenetic changes associated with autoimmune diseases may be influenced by the sex or by external factors. Genetic imprinting, especially microRNA (miRNA) one, might also play a role in the sexual dimorphism observed in autoimmune diseases [88].

## 4. Gut Microbiota, Gender, and Vaccines

The term “microbiota” refers to the amount of microbes colonizing the human body in various locations and living in symbiosis with the host in a condition of health [90]. Through the expression of more than 3 million genes, the gut microbiota (GM) plays a pivotal role in physiological mechanisms. Conversely, its deregulation has been linked to pathological processes affecting several organs and apparatuses [91]. The GM composition is the result of an intense interplay between genetic and environmental factors, such as age, sex, perinatal feeding, dietary habits, lifestyle and antibiotic exposure [92].

Sex is one of the most important factors influencing the gut microbiota, since men and women diverge in microbial composition and diversity from birth. For instance, male newborns have a lower alpha-diversity, a lower abundance of Clostridiales, and a higher abundance of Enterobacteriales than females in early life [93]. Interestingly, Nagpal et al. described a higher abundance of Bifidobacteria on the first day of life in male infants compared to females [94]. Bifidobacterial species are the main components in the GM of breastfed infants; in particular, *Bifidobacterium (B.) breve*, *B. longum subspecies longum*, *B. longum subspecies infantis*, and *B. bifidum* have been identified in their feces [95].

The GM undergoes gradual transformations during childhood (e.g., increase in microbial stability and development of specific Clostridium clusters) [96] without significant sex differences until puberty. An analysis of the GM composition in 13-year-old adolescents showed a shift towards an adult profile with pubertal progression only in girls, with an increase in Clostridia (e.g., Ruminococcaceae) with estrogen-metabolizing activity [97]. Yuan et al. excluded sex differences in alpha- and beta-diversity in pre-puberty, compared to pubertal subjects, who showed variances at the genera level (e.g., prevalence of *Lactococcus* and *Rothia* in boys, and prevalence of *Alistipes* and *Oscillospira* in girls) [98].

Sex hormones play a crucial role in shaping the sexual dimorphism of the GM. Estrogen levels are positively correlated to alpha-diversity, a lower abundance of Firmicutes, and a higher abundance of Bacteroidetes and Clostridiales, mostly the Ruminococcaceae family. On the contrary, testosterone increases *Ruminococcus*, *Acinetobacter*, and GM diversity in men, and *Shigella* and *Escherichia* in women [99]. The microbiota can also impact steroid metabolism by reactivating estrogens, deactivating androgens, and modifying sex hormone levels [100]. The interplay between the microbiome and estrogens—through which gut microbes metabolize estrogens and, in turn, are influenced by the estrogenic metabolites—has been defined as the “estrobolome” [101].

Ageing is accompanied by physiological changes in systemic functions, together with an increased burden of disease, often regarding the gastrointestinal tract and a higher use of medications that may affect the GM status [102]. In the elderly, a reduction in sex microbial differences in the GM (e.g., alpha-diversity) has been observed [103], probably due to lower levels of sex hormones, particularly in postmenopausal women. Flores et al. revealed a positive correlation between the levels of urinary estrogens and the richness of the GM in both men and postmenopausal women [104]. Taken together, these data confirm the role of sex hormones in shaping the GM composition throughout human life.

Sex differences in GM could explain, in certain cases, the existence of unequal predisposition to disease between the two sexes. For instance, women with metabolic syndrome have an increased amount of Alistipes, Collinsella, and Phascolarctobacterium genera, while men an increased amount of *Prevotella* and *Faecalibacterium*, suggesting the role of the GM in the different incidences of metabolic diseases [105].

The gut microbiota takes part in a dynamic crosstalk with the host immune system, since the microbial metabolites influence the relationship between intestinal epithelial cells and immune cells [106]. Indeed, the GM species and their metabolites (e.g., short-chain fatty acids) have a regulatory function in immune cell activation and differentiation [107]. In the gut, the existence of an intact mucosal barrier is essential to prevent the potential damage induced by toxicants or pathogenic bacteria [108]. Gut dysbiosis and the associated disruption of the epithelial barrier can induce immune dysregulation, inflammation, oxidative stress, and metabolic disorders [106].

In the last few years, given its strong connection with both innate and adaptive immunity, the GM has been evaluated as a determinant of the immune response to vaccination. The GM influences vaccine immunogenicity through various mechanisms, such as cross-reactive epitopes between microorganisms and vaccines, the regulation of the B cell production of antibodies, microbial-derived natural adjuvants stimulating pattern recognition receptors, and the regulation of the plasmacytoid dendritic cell production of type I interferon [109].

Intestinal immune responses differ between sexes since males have a lower amount of T cells—with the exception of Th1 cells—and a higher amount of CD80+ DCs and NK cells in the Peyer’s patches [110]. For this reason, at the gut level, male mice seem to build stronger innate immune responses and weaker adaptive immune responses, as compared with females. However, another study conducted on rats showed higher macrophage (CD68+) populations within the mesenteric lymph node in females compared to males [111].

Fransen et al. tried to identify sex-specific immune differences associated with the GM composition through the GM transfer from male or female mice to germ-free mice. They demonstrated that some immune differences, such as the enhanced type I IFN pathway in the female gut, were microbiota-independent. This sexual dimorphism is possibly responsible for the selection of a sex-specific GM with an overgrowth of *Alistipes*, *Rikenella*, and Porphyromonadaceae in males [112].

The role of the gut microbiota in the immune sex bias has been extensively investigated in the field of autoimmune diseases. For instance, non-obese diabetic female mice develop type 1 diabetes more frequently than males; this is not observed in germ-free mice [113]. In this context, steroid hormones play a crucial role, since the GM of castrated male mice is similar to the female microbiota and the male incidence of autoimmune diseases increases after androgen depletion [113]. Moreover, in a murine model of primary biliary cholangitis (PBC) treated with antibiotics, a reduction in sex differences in lymphocytic infiltration and inflammatory cytokines has been demonstrated, thus revealing a GM-mediated sex bias in PBC [114].

The GM composition is responsible for variations in immune responses to vaccines against respiratory pathogens. A clinical study by Nakaya et al. described the positive correlation existing between early TLR5 expression after vaccination and the hemagglutination-inhibition (HAI) titers dosed at 4 weeks [115]. Accordingly, TLR5⁻^/^⁻ mice develop reduced antibody titers after influenza vaccination. Interestingly, the oral administration of flagellated—but not aflagellated—*E. coli* enhanced humoral responses, due to the role of flagellin as a TLR5 ligand stimulating plasma cell differentiation directly or by inducing the macrophage production of growth factors [116].

The gut microbiota influences the immunogenicity of the COVID-19 mRNA vaccines. A prospective cohort study found a positive correlation between pre-vaccination microbial diversity and composition (e.g., the phylum Desulfobacterota and genus *Bilophila*, synthesizing immunostimulatory endotoxin) and the final levels of anti-spike IgG [117]. Moreover, the antibiotic-induced depletion of the gut microbiome in mice was observed to reduce the immunogenicity of the Pfizer (BNT162b2) vaccine [117].

In a novel mouse study, Amato-Menker et al. have found that the immune response to heat-killed *Streptococcus pneumoniae* immunization is modulated by the gut microbiome in a sex chromosome complement-dependent manner. In particular, XX mice exhibit higher frequencies of IgM-secreting B cells and plasma cells than XY mice, as well as an overexpression of the *Kdm6a* gene, located on the X chromosome. Interestingly, in XX mice, microbiota depletion impairs the humoral responses, which can be, in turn, reconstituted by the administration of short-chain fatty acid-producing bacteria [118]. These data suggest a potential correlation between gut microbiota and sexual dimorphic vaccine responses.

Understanding the immunological interplay between sex, vaccines, and microbiota could make the latter a potential target of novel approaches to improve vaccine immunogenicity [119]. In the future, bioengineering approaches targeting the GM in a sex-specific manner may increase the immune responses to vaccines. For example, a murine study showed that an oral recombinant yeast probiotic exhibiting the SARS-CoV-2 spike protein could both activate the immune system and modify the GM composition with sex differences (Succinivibronaceae, Atopobiaceae, and Akkermansiaceae in male mice, and Desulfovibrionaceae in female mice) [120].

## 5. Sex Dimorphisms in Immune Response to Vaccines

Vaccination provides protection against pathogens through mechanisms involving both the innate and the adaptive immune systems with the aim of achieving durable immunity [121].

Sex differences exist even in the first-line immune responses. For example, women produce higher levels of IFN-alpha, while men higher levels of the immunosuppressive cytokine IL-10, in response to TLR7 ligand stimulation [122]. After a trivalent influenza vaccination, greater amounts of inflammatory cytokines, in correlation with the levels of monocyte phosphorylated STAT3, were detected in women compared to men regardless of age. However, sex differences in leptin, C-reactive protein, and IL-1 receptor agonist levels were less evident in the elderly due to the increased serum concentration of these proteins in older men compared to young men [123]. Female mice’s antigen-presenting cells (APCs) are possibly more efficient than those in males due to a higher expression of major histocompatibility complex (MHC) class II and co-stimulatory molecules [124]. In addition, animal models have revealed a sexual dimorphism in innate immunity after yellow fever virus (YFV) vaccination, due to an overexpression of genes associated with TLR pathways and IFN responses in females [125]. In the field of adaptive immunity, females generally have higher basal levels of immunoglobulins and develop stronger humoral responses to vaccination. In fact, after vaccination against hepatitis A and B, herpes simplex virus type 2, rabies, smallpox, and dengue viruses, antibody titers can be two times higher in females compared to males of all ages [125,126]. However, in women, a more rapid decline of antibodies has been shown in certain cases, for instance, after hepatitis A immunization [127].

As for antibody subclasses, several studies have documented a more robust production of functional and inflammatory immunoglobulins in female mice. In fact, after the immunization with a trivalent influenza vaccine, female mice showed higher levels of IgM and a higher IgG2a/IgG1 ratio, as well as a shift toward Th1 in the Th1/Th2 balance [128]. In humans and mice, the upregulation of IgG2 depends on the binding of the estrogen receptor to estrogen response elements and cytosine-adenine repeats upstream of constant region (C) gamma genes within the immunoglobulin heavy chain loci [129]. Additionally, after pertussis vaccination, men produce higher titers of poorly effective specific IgG4 compared to women, with a greater occurrence of IgG4 in the youngest children and a gradual decline with age [130].

Another aspect is the impact on the immune responses of immunoglobulin glycosylation, since glycan composition, which depends on sex hormone plasma levels, influences the affinity of IgG to ligands [131]. Different patterns of glycosylation for the IgG subclasses have been described in humans after meningococcal, pneumococcal, and influenza vaccination [132]. Interestingly, with ageing, the reduction in sex hormones in women promotes antibody galactosylation [133], resulting in less inflammatory antibodies, as well as N-glycan-branching in T cells, with the consequence of a negative regulation of T cell activity [134].

In a mouse model of influenza vaccination, the passive transfer of immunoglobulins from vaccinated females into naïve mice led to more robust protection by reducing the virus titers in lungs. The authors described a different gene expression following vaccination in the two sexes, with a higher expression of TLR7, located on the X chromosome, in female mice due to epigenetic changes (reduced DNA methylation in females) in the promoter region [135]. TLR7 is an innate immune receptor that also plays a crucial role in the Ig class switch DNA recombination [136]. Moreover, females develop a higher number of CD8+ T memory cells after influenza infection, but males and females are equally protected against reinfection. The amount of CD8+ memory T cells was higher following the infection rather than following vaccination in both sexes. In particular, tissue resident memory (TRM) cells provide significant protection through their rapid expansion in the lungs and their ability to kill infected cells, engaging circulating memory T cells and producing cytokines [135].

In some countries, a lower measles-specific antibody-dependent cellular cytotoxicity (ADCC), rather than the neutralizing antibody activity, seems to correlate with reduced survival rates in young women [137].

Cell-mediated immune responses (e.g., T-cell activation) to some vaccines are stronger in women than men [123,138,139]. Among the indicators of cell-mediated immunity, mitogen-stimulated lymphocyte proliferation, immunological intolerance to non-self, and wound healing appear higher in women than in men [126].

The flow cytometry evaluation of peripheral blood cells revealed a greater amount of CD4+ T cells in women compared to men; moreover, the individuals with an inversion of CD4/CD8 ratio were found to be predominantly men with an increased prevalence at an older age [140]. In a murine model of infection, Yee Mon et al. demonstrated that CD8+ T cells become more frequently short-lived effectors in females due to an enhanced response to IL-12, whereas they become memory precursor effectors in males [141]. In general, women develop stronger cytotoxic T-lymphocyte responses compared to men, as well as an upregulation of important inflammatory/cytotoxic effector genes, such as lymphotoxin beta, granzyme A, granulysin, and IFN-gamma, as revealed by microarray analysis in healthy individuals aged 25–35 [142]. Additionally, females exhibit a higher number of regulatory T cells (Tregs) [143], involved in immune tolerance in pregnancy, as confirmed in mouse models [144].

To summarize, several biological mechanisms have been proposed for the gender-based differences in immune responses to vaccination [145].

As discussed above, sex steroids could influence the function of immune cells. Sex steroids bind to specific receptors expressed in lymphoid tissue cells but also in circulating lymphocytes, macrophages, and dendritic cells [146]. Age-related deregulation of the immune system depends on a reduction in sex hormone concentrations and receptor signaling with ageing [147]. However, it is noteworthy that hormonal replacement therapies do not modify the outcome of vaccination [148].

Genetic and epigenetic factors contribute to sex differences in the immune response to vaccines [149,150]. Numerous genes encoding proteins involved in immune responses are situated on the X chromosome, as well as genes encoding effectors of transcriptional and translational control, downstream of activated cytokine receptors [151]. Polymorphisms or mutations of X-linked genes have more serious immune consequences in men than in women [152,153].

As previously analyzed, hormonal status influences gut microbiome composition, resulting in sex-specific microbiome profiles [154]. In particular, higher levels of estrogen promote the richness and the diversity of the gut microbiome [155]. Moreover, male mice transplanted with female microbiota had a higher number of T-cell precursors in the thymus, as well as a reduction in anti-inflammatory cells [112]. These results suggest that female hormones improve the immune response to infections through a pro-inflammatory effect [79]. Furthermore, the immune function and the development of autoimmune disorders, such as type 1 diabetes, are mediated by the sex-specific composition of the gut microbiome and the hormonal status [113,156,157].

Latent infections can also affect the outcome of vaccination. Cytomegalovirus (CMV) latency, in particular, is associated with elevated antibody responses to influenza vaccines in young adults [158]. In the elderly, CMV seropositivity is associated with chronic inflammation, a decreased amount of CD4^+^ T cells that are also less responsive to neoantigens [159], and lower humoral responses to influenza vaccination [160,161,162]. Moreover, latent CMV infection reduces vaccine effectiveness against severe acute respiratory syndrome coronavirus 2 (SARS-CoV-2) infection in terms of immunoglobulin anti-spike protein and anti-receptor binding domain [163]. Other chronic infectious diseases are associated with an impaired immune response to vaccination. In fact, hepatitis B virus infection leads to less protective response against tetanus, and human immunodeficiency virus (HIV) infection is linked to poorer humoral responses to hepatitis A, hepatitis B, and measles vaccines. The possible existence of sex differences is not clear yet [164].

## 6. Sex Differences in Vaccine Outcome

In the scientific literature, several human and animal studies have investigated the impact of sex on vaccine responses.

In a seronegative population, a significant immunological parameter expressing antibody response after vaccination is the post-vaccination geometric mean titer (post-GMT). The related seroprotection rate (SPR) is a valid parameter to express the protection that an administered vaccine provides [165].

Human studies focusing on sex differences in response to the measles–mumps–rubella (MMR) vaccine led to inconsistent findings. Some of them describe higher GMTs or SPRs after MMR vaccination in women [166,167,168], while other studies report transiently higher GMTs in men or no sex differences [169,170].

In particular, a Spanish study including children and adults aged 15 years or over, who had been vaccinated against MMR, revealed a higher prevalence of protective antibodies in female individuals than males [171]. Another study reported that antibodies against rubella virus and lymphocyte proliferation were transitorily higher in boys at weeks 2 and 4 following vaccination, with no sex difference 10 weeks after MMR vaccination [169]. In addition, girls have a greater long-term protection against rubella compared to boys, as observed by dosing the specific antibody titers at 14–17 years of age [172]. However, boys seem to develop higher levels of IL-6, IL-1beta, interferon (IFN)gamma, and TNFalpha after MMR vaccination. In a cohort of 748 individuals with an average age of 14.9 years, these results were observed in the older participants, suggesting the role of puberty in immune outcomes [173].

Ohm and colleagues, investigating the immune response to meningococcal vaccination in adolescents, found that girls had higher antibody responses compared with boys, although almost all the vaccinated subjects reached protective titers [174].

Sex differences in the humoral immune responses to yellow fever virus (YFV) vaccination have not been reported. Interestingly, a few days after YF vaccination in adult volunteers, the expression of over 600 genes changed in women, while only 67 genes were encoded differently in men [10,125,126]. These differentially expressed genes are TLR and IFN-associated genes, involved in the early innate immune response. It is not clear if the efficacy of the YFV vaccine is higher in women compared to men [175,176].

The effectiveness of the Bacillus Calmette–Guerin (BCG) vaccine, recommended for infants in tuberculosis-endemic countries, is a matter of debate, since it does not provide protection against tuberculosis in older children or adults without booster doses [177]. Some studies show that non-specific effects of the BCG vaccination on survival and a reduction in respiratory infections are greater in girls than boys [178,179]. The results of one trial evaluating the impact of BCG vaccination on all-cause mortality at 71-year follow-up confirmed its protective non-specific effects, especially in women [180]. The real changes induced by the vaccine in immune function to obtain these positive effects have not been clarified. On the other hand, BCG vaccination has been observed to decrease systemic inflammation in an adult population with a median age of 26 years, with a stronger effect seen in men than in women [181].

Influenza vaccines, in spite of periodical reformulation, can fail in preventing disease due to both poor vaccination coverage [182] and low vaccine effectiveness (effectiveness 54% in flu season 2022–2023, in contrast to vaccines for other viral diseases, such as the measles vaccine which is effective in 97% or the mumps vaccine which is effective in 88%) [183,184]. The main barrier to vaccine success is antigenic variation, but age and sex are other important determinants.

A retrospective study by Sánchez-de Prada et al., based on a serological analysis of more than two thousand subjects just before and one month after the influenza vaccination, showed a higher seroconversion rate against some viral types in elderly women, differently from young adults [185].

Yang et al., in a study on elderly people, detected the presence of differentially expressed genes (DEGs) involved in sex-specific immunity to quadrivalent inactivated influenza vaccines. In particular, in women, an upregulation of DEGs associated with type I interferon responses and complement activation was observed [186].

Taken together, both preclinical and clinical studies confirm a greater influenza vaccine immunogenicity in women, since they have more efficient germinal center B cells and develop a higher humoral response [187,188,189].

Sex differences are also evident in the development of adverse effects to vaccines [190]. For example, in studies about the safety of influenza vaccines, women generally report adverse effects following immunization (AEFIs) more frequently than men. AEFIs can be local (e.g., site injection pain/swelling/redness) or systemic (e.g., fever). The burden of adverse effects seems to contribute to the increased vaccine hesitancy observed in women [189].

Vaccines can trigger autoimmune reactions within the recipient’s immune system, although these occurrences are less commonly reported compared to the more typical transient, acute side effects.

Also, an “autoimmune/inflammatory syndrome induced by adjuvants” (ASIA) has been described. This encompasses a set of immune reactions caused by certain compounds found in vaccines, such as silicone, aluminum, or infectious components [191,192]. Watad et al. studied 500 patients aged 43 ± 17 years with ASIA syndrome and found that 89% were women, thus suggesting a higher risk of developing this condition compared to men [193,194].

As previously discussed, women have a higher expression of X-linked genes encoding for cytokines, chemokines, and cell surface immune markers [39]. Furthermore, women typically demonstrate a higher neutrophil count than men in various inflammatory conditions and elevated baseline erythrocyte sedimentation rate levels [28]. Additionally, after vaccination, adult women tend to develop greater IL-6 and antibody responses and those correlate with estradiol concentrations [195]. Consequently, women demonstrate heightened immune system responsiveness in contrast to men [196], which may account for the increased incidence of side effects observed after vaccination.

A clinical trial by Engler and colleagues revealed that the humoral response to a half dose of trivalent inactivated influenza vaccine in adult women is comparable to the response to a complete dose in men [197], suggesting that reducing the vaccine dosage may be advantageous in women.

Adverse events following 17D vaccination for yellow fever are considerable in public health. An analysis of adverse effects following YF vaccination, reported by the U.S. Vaccine Adverse Event Reporting System, showed that the majority of serious side effects, anaphylaxis, neurologic, or viscerotropic disease occurred in males, particularly at an older age [198].

Several studies have recently focused on COVID-19 vaccines and the adverse effects following vaccination that can be mild or more dangerous, such as anaphylaxis [199,200].

Green et al. reported adverse effects after two or three doses of the Pfizer-BioNTech COVID-19 vaccination, showing that the risk of local events (e.g., injection-site soreness), systemic events (e.g., fever), and sensory events (e.g., paresthesia) was significantly higher in women independently of age [201].

Yin et al. conducted a prospective study on adverse events following influenza or COVID-19 vaccination in adult healthcare workers [202]. At any age, women reported adverse events following both influenza and COVID-19 vaccination more frequently than men [202].

Few cases of myocarditis and pericarditis have been described after the administration of COVID-19 mRNA vaccines. Le Vu et al., analyzing patients with myocarditis and pericarditis following vaccination, found an increased risk of cardiac injury during the first week after the receipt of vaccine. The stronger association was reported for myocarditis subsequent to mRNA-1273 vaccination in both sexes aged 18–24 [203]. A systematic review conducted by Ling et al. showed a low risk of myopericarditis following COVID-19 vaccination. However, a higher incidence was observed in boys, especially after mRNA vaccines [204].

A meta-analysis observed a correlation between COVID-19 vaccine and menstrual irregularities; specifically, menorrhagia, oligomenorrhea, and polymenorrhea were observed as the most frequent events [205]. While COVID-19 vaccines seem to have an impact on women’s menstrual cycles, the effects are usually temporary and well tolerated [206]. A study based on two large cohort populations observed an increased risk of unexpected vaginal bleeding after COVID-19 vaccination among peri- and postmenopausal women, possibly due to changes in the endometrium induced by a local response to the spike protein. A two- to three-fold increase in risk was observed in postmenopausal women in the 4 weeks after vaccination compared to the period before vaccination [207].

Studies on sex-based differences in the efficacy of COVID-19 vaccination need to be designed. A meta-analysis described a significantly increased efficacy in men compared to women in the vaccine group. Men had a 33% decrease in the overall risk of COVID-19 after vaccination compared to women. However, the real impact of sex on the outcome of COVID-19 vaccines should be evaluated in a larger number of studies, particularly in clinical trials [208,209].

The human papillomavirus (HPV) vaccine is an example showing a different acceptability of vaccination across the sexes.

HPV is a DNA virus and one of the most common causes of sexually transmitted disease. The oncogenic “high-risk” genotypes of HPV are responsible for anogenital cancers and cancers of the head and neck. In industrialized countries, the impact of HPV-related diseases in men is equivalent to that in women. In this context, since HPV vaccination has been proven to be safe and useful in preventing genital warts and cancers also in men, health systems have begun promoting male vaccination [210].

A meta-analysis by Newman et al. shows a partial acceptability of the HPV vaccine among men [211]. In contrast, acceptability was considerably higher in a review of US studies focusing on young women [212]. Vaccine acceptability is inextricably linked to the individual perception of vaccine benefits and the recommendations by health professionals. For this reason, public health campaigns providing awareness about HPV risk in men and recommending vaccination, as well as strategies to reduce logistical and economic impediments to vaccine intake, should be designed [211] (Table 2).

## 7. Conclusions

As clearly shown during the recent COVID-19 pandemic, sex is part of an intense crosstalk with other factors influencing the natural history of infections and the outcomes of vaccination [214].

The differential immune modulation in men and women, resulting from hormonal status as well as from genetic and epigenetic variables, is responsible for the sex-specific responses to vaccines.

In the past decades, most vaccine studies, both preclinical and clinical, have not focused on sex-related differences, hypothesizing that their results could be considered generalizable to the entire population.

However, men and women have different immunological responses, and they should be equally represented in future vaccine trials. Moreover, the study outcomes should be measured with a sexual perspective in order to increase vaccine efficacy and minimize adverse effects, leading to personalized vaccinology [215].

Another important aspect that requires further research is the impact of the gut microbiota on vaccine efficacy in men and women throughout life. Ten years ago, Flak and colleagues described the concept of the “microgenderome” [216], referring to the relationship between the microbiota, sex hormones, and immune system. Sex-related differences in gut microbiota composition have been linked to changes in gut and systemic inflammation, immune functions, and the development of inflammatory diseases [217]. However, there is still a lack of evidence about their effects on the immune responses to vaccines.

Moreover, a significant issue that modern vaccinology will need to address is the rising age of the population and the consequent immunosenescence. Immunosenescence affects the function of both innate and adaptive immune cells, hindering the efficient generation of memory lymphocytes, resulting in decreased antibody response and accelerated decline in antibody titers among the elderly [218]. Various approaches can be adopted to improve vaccine responses in the elderly, such as enhanced vaccine formulations, higher antigen concentrations, more immunogenic adjuvants, booster injections, appropriate methods and timing for vaccine delivery, modified immunization schedules, vaccine routes of administration, and immunomodulators [8].

The possible correlation between immune-mediated diseases and vaccines could lead, in the future, to the development of novel treatments, also for cancer. In particular, the aim would be to target the neoplastic stem cells, the drug-resistant populations, or the epithelial–mesenchymal transition phenotype, in combination with other immune-based therapies (e.g., checkpoint inhibitors) and traditional therapies [219].

Interestingly, some cancer-specific antigens are expressed differently between the two sexes. For instance, the expression of NY-SAR-35 and MAGE3, associated with lung cancer, is higher in the male sex, whereas Ropporin-1, associated with multiple myeloma, is higher in the female sex. This implies the need for a sex-oriented approach in the immunotherapeutic development of novel cancer vaccines [220].

We are also looking in the direction of tertiary prevention to reduce the risk of post-treatment recurrence, even if, to date, the progress of vaccines has been mainly in the prevention of cancer, such as carcinoma of the uterine cervix in women and hepatocellular carcinoma caused by oncogenic viruses [221].

Finally, modern vaccinology cannot underestimate the psychological dimension of vaccination. It is important to analyze the reasons behind vaccine hesitancy, which is often sex-related, in order to increase awareness about the benefits and real risks of vaccination [222].

In this context, educational campaigns about vaccines should be sex-specific and public institutions should indicate both vaccine schedules and doses specifically for women and for men.

## Figures and Tables

**Figure 1 cells-13-00526-f001:**
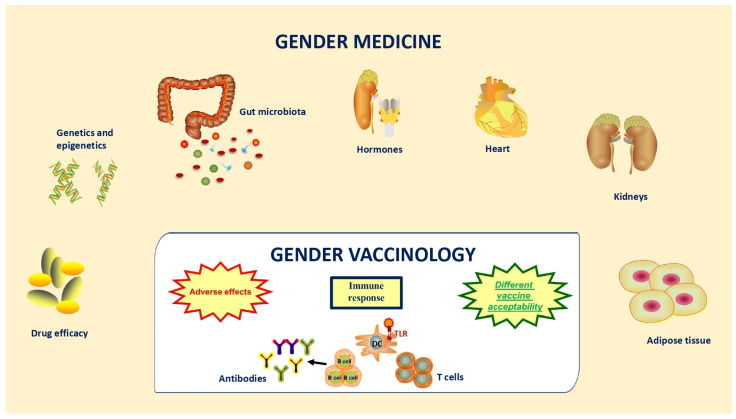
Since the first definition of “gender medicine” came out in the early nineties, the scientific community has investigated the impact of sex on physiology, physiopathology, and clinical features of disease in order to provide evidence-based therapeutic decisions. A novel appealing chapter within gender medicine is “gender vaccinology”, which explores the benefits and risks of vaccination in both men and women with the aim of encouraging strategies of health promotion.

**Table 1 cells-13-00526-t001:** The effects of steroid hormones on the immune system.

Steroid Hormones	Immune Cell Types and Molecules	Effect	References
Estrogen	IL-10	Increased secretion	[14]
	Dendritic cells	Regulatory role	[14]
	CD4^+^ T cells	Immunoactivating effect	[75]
	B cells	Boosting humoral responses, promotion of the differentiation and the production of Igs	[79]
Progesterone	Dendritic cells	Immunosuppressive effect	[60]
	Macrophages	Immunosuppressive effect	[60]
	NK cells	Immunosuppressive effect	[60]
	CD4^+^ T cells	Immunosuppressive effect	[75]
Testosterone	Neutrophil	Promotion of differentiation and recruitment	[69]
	Macrophage	Immunosuppressive effect	[66]
	Dendritic cells	Immunosuppressive effect	[72]
	Th1 cells	Immunosuppressive effect	[76]
	B cells	Immunosuppressive effect	[77]

Abbreviations: IL, interleukin; CD, cluster of differentiation; Igs, immunoglobulins; NK, natural killer; Th1, T helper 1.

**Table 2 cells-13-00526-t002:** The roots of sex differences in vaccine outcome.

Field	Features	Potential Determinants	References
Immune protective response	Antibody titers Lymphocyte proliferation and activation	Sex steroids Genetic factors Epigenetic factors Gut microbiome Latent infections	[125,126,145]
Adverse effects	Local or systemic Mild or severe	Different immune activation Inflammation	[126,189]
Vaccine acceptability	Greater vaccine hesitancy in women Reduced acceptability of HPV vaccination in men	Cultural norms Social barriers Adverse events following vaccination Psychological dimension	[190,211,213]

Abbreviations: HPV, human papillomavirus.

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
