# Peer review of "Immune Cells, Gut Microbiota, and Vaccines: A Gender Perspective"

_cells, 2024, doi:10.3390/cells13060526_

Round 1

Reviewer 1 Report

Comments and Suggestions for Authors

This review describes the gender differences in the immune response to vaccines by comparing data on immune responses, vaccine efficacy, and side effects. The article also shows that considering the gender perspective can improve the efficacy of the immune response and enable the development of customized vaccination regimens.

This is an interesting topic that has been discussed for a long time, but little is known about it, especially about the microbiome and its composition depending on a person's gender and its impact on the regulation of the immune response. In addition, the mechanisms underlying the greater hyperreactivity of the immune system in women are not yet sufficiently understood. They may be the reason why adverse reactions to vaccinations, rheumatic diseases, and autoimmune diseases are more common in women.

The article deserves to be published by reviewing the conclusions. Certainly, modern vaccinology needs to improve information to overcome hesitancy, but it also needs to work on developing gender- and age-appropriate adjuvants. Decreasing susceptibility to vaccines with age was only mentioned in passing, but is certainly another issue that the vaccine industry should work on.

Author Response

Rome, 10th March, 2024

Dear Editor of “Cells”,
first of all, my coauthors and I would like to thank You sincerely for this opportunity to cooperate, following the submission of the paper “Immune cells, gut microbiota, and vaccines: a gender perspective” (Manuscript ID: cells-2899032) and its possible publication upon “Cells”.
We profoundly thank the reviewers for the comments and useful suggestions aimed at improving the final version of the paper. 
This is a point-by-point list of changes made in the paper:

REVIEWER 1
This review describes the gender differences in the immune response to vaccines by comparing data on immune responses, vaccine efficacy, and side effects. The article also shows that considering the gender perspective can improve the efficacy of the immune response and enable the development of customized vaccination regimens.
This is an interesting topic that has been discussed for a long time, but little is known about it, especially about the microbiome and its composition depending on a person's gender and its impact on the regulation of the immune response. In addition, the mechanisms underlying the greater hyperreactivity of the immune system in women are not yet sufficiently understood. They may be the reason why adverse reactions to vaccinations, rheumatic diseases, and autoimmune diseases are more common in women.
The article deserves to be published by reviewing the conclusions. Certainly, modern vaccinology needs to improve information to overcome hesitancy, but it also needs to work on developing gender- and age-appropriate adjuvants. Decreasing susceptibility to vaccines with age was only mentioned in passing, but is certainly another issue that the vaccine industry should work on.

Thank you for your suggestion.

In the conclusions, we have included some sentences about vaccines in older individuals, suggesting potential future approaches for effective vaccines considering immunosenescence.

We thank You for your constructive critique and we hope the review process has led to an improved manuscript.
If additional changes are warranted, we will make them.
We hope that this revised version of our manuscript may now be found suitable for publication.
Sincerely, 
Rossella Cianci, MD, PhD

Reviewer 2 Report

Comments and Suggestions for Authors

In this review, authors have described in details about what we know related to biological sex and gender perspectives during vaccination and why this is relevant to design tailored vaccine formulations. It is written nicely. My minor suggestions would be to include more articles exploring sex differences in innate and adaptive immune responses to HIV, influenza, and SARS-CoV-2 from Drs. Marcus Altfeld, Sabra Klein, and Akiko Iwasaki's lab. These are the labs that regularly study sex differences during infectious diseases and vaccines and several relevant articles are missing to give a comprehensive idea of what we already know. Otherwise, looks good. 

Author Response

Rome, 10th March, 2024

Dear Editor of “Cells”,
first of all, my coauthors and I would like to thank You sincerely for this opportunity to cooperate, following the submission of the paper “Immune cells, gut microbiota, and vaccines: a gender perspective” (Manuscript ID: cells-2899032) and its possible publication upon “Cells”.
We profoundly thank the reviewers for the comments and useful suggestions aimed at improving the final version of the paper. 
This is a point-by-point list of changes made in the paper:

REVIEWER 2
In this review, authors have described in details about what we know related to biological sex and gender perspectives during vaccination and why this is relevant to design tailored vaccine formulations. It is written nicely. My minor suggestions would be to include more articles exploring sex differences in innate and adaptive immune responses to HIV, influenza, and SARS-CoV-2 from Drs. Marcus Altfeld, Sabra Klein, and Akiko Iwasaki's lab. These are the labs that regularly study sex differences during infectious diseases and vaccines and several relevant articles are missing to give a comprehensive idea of what we already know. Otherwise, looks good. 

Thank you for your suggestion. We have included some works by the authors suggested.

We thank You for your constructive critique and we hope the review process has led to an improved manuscript.
If additional changes are warranted, we will make them.
We hope that this revised version of our manuscript may now be found suitable for publication.
Sincerely, 
Rossella Cianci, MD, PhD

Reviewer 3 Report

Comments and Suggestions for Authors

the manuscript tries to combine Immune cells, gut microbiota, and vaccines over gender too. This manuscript reminded us of the saying “The more you can combine the seemingly unfamiliar, the more you will get.”  many appreciated messages authors released, one interesting and important is that mentioned thoroughly in the MS and concisely in lines 765-768, that we do not care enough during COVID-19 Vaccines (especially the new platforms) clinical trials. The manuscript is acceptable in the current version, however, it is highly recommended to add the responses for the following comment in the new version to increase its impact.

1.      Word corresponding author changes from superscript to normal.

2.      Your manuscript title and keywords contain “gut microbiota” so, must be implemented it in the right and organized position in the abstract and introduction too. Then link gut-microbiota with all issues mentioned in your introduction, which will demonstrate to readers the impact roles microbiota plays.

3.      Lines 13-14, as we know that there are preventive and therapeutic vaccines, so please write in this sentence in light this fact.

4.      Vaccines did and still a huge for humanities, so please update the data in lines 28-30 using these links: https://www.who.int/health-topics/vaccines-and-immunization#tab=tab_1 ; https://www.cdc.gov/globalhealth/immunization/data/fast-facts.html

5.      Please try to discriminate –during your rephrases-between adult men/women and pediatric of both sexes because there are differences in all issues you release and compare.

Author Response

Rome, 10th March, 2024

Dear Editor of “Cells”,
first of all, my coauthors and I would like to thank You sincerely for this opportunity to cooperate, following the submission of the paper “Immune cells, gut microbiota, and vaccines: a gender perspective” (Manuscript ID: cells-2899032) and its possible publication upon “Cells”.
We profoundly thank the reviewers for the comments and useful suggestions aimed at improving the final version of the paper. 
This is a point-by-point list of changes made in the paper:

REVIEWER 3
The manuscript tries to combine Immune cells, gut microbiota, and vaccines over gender too. This manuscript reminded us of the saying “The more you can combine the seemingly unfamiliar, the more you will get.”  many appreciated messages authors released, one interesting and important is that mentioned thoroughly in the MS and concisely in lines 765-768, that we do not care enough during COVID-19 Vaccines (especially the new platforms) clinical trials. The manuscript is acceptable in the current version, however, it is highly recommended to add the responses for the following comment in the new version to increase its impact.

1.      Word corresponding author changes from superscript to normal.

We corrected this error.

2.      Your manuscript title and keywords contain “gut microbiota” so, must be implemented it in the right and organized position in the abstract and introduction too. Then link gut-microbiota with all issues mentioned in your introduction, which will demonstrate to readers the impact roles microbiota plays.

Thank You for your suggestion. As requested, we implemented the abstract and the introduction by mentioning the gut-microbiota and its link with the topics of this paper.

3.      Lines 13-14, as we know that there are preventive and therapeutic vaccines, so please write in this sentence in light this fact.

Thank You for your suggestion. We specified the existence of both preventive and therapeutic vaccines.

4.      Vaccines did and still a huge for humanities, so please update the data in lines 28-30 using these links: https://www.who.int/health-topics/vaccines-and-immunization#tab=tab_1; https://www.cdc.gov/globalhealth/immunization/data/fast-facts.html

Thank You for your advice. As requested, we updated the data using the indicated references.

5.      Please try to discriminate –during your rephrases-between adult men/women and pediatric of both sexes because there are differences in all issues you release and compare.

Thank You for your suggestion. We discriminated throughout the text, where not clear, between adult and pediatric individuals and specified the age of men and women included in the reported studies.

We thank You for your constructive critique and we hope the review process has led to an improved manuscript.
If additional changes are warranted, we will make them.
We hope that this revised version of our manuscript may now be found suitable for publication.
Sincerely, 
Rossella Cianci, MD, PhD

Round 2

Reviewer 3 Report

Comments and Suggestions for Authors

Thank you for your positive reply.